# A Study of the Emotional Intelligence and Personality Traits of University Finance Students

Ana M. Rosales-Pérez [1,*], Manuel A. Fernández-Gámez [2], Macarena Torroba-Díaz [1] and Jesús Molina-Gómez [3]

[1] MAINAKE Research Team, University of Malaga, 29016 Málaga, Spain; macarenatorrobadiaz@uma.es
[2] Department of Finance and Accounting, University of Malaga, 29016 Málaga, Spain; mangel@uma.es
[3] Department of Economics and Business Administration, University of Malaga, 29016 Málaga, Spain; jmolinag@uma.es
[*] Correspondence: anamrp@uma.es; Tel.: +34-615-894-356

**Abstract:** Studies on financial behavior indicate that emotional intelligence (EI) and personality traits (PTs) explain much of the bias in financial activity. This study aims to identify in which dimensions of theEI and PTs of university students in finance further training is needed to avoid financial behavior bias. To this end, the EI and PT levels of a sample of university finance students and financial industry professionals were compared using the Trait Emotional Intelligence Questionnaire (TEIQue) and Big Five Inventory questionnaire. Subsequently, the dimensions of EI and PTs in which students have a deficit compared to professionals were identified, and the impacts that this deficit causes on the financial behavior of students were determined. The results indicate that students are deficient in the EI competencies related to empathy, emotion regulation, self-motivation, stress management, optimism, and self-esteem. Furthermore, PTs are related to kindness, awareness, openness, and extraversion. This deficit makes students more likely to have financial behavior biases such as risk tolerance, endowment, optimism, self-control, and loss aversion. These findings suggest that universities should be aware of providing financial students with full training in EI and PTs to help them successfully address their professional future.

**Keywords:** financial education; emotional intelligence; personality traits; financial behavior; university students

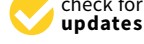

## 1. Introduction

Studies on financial behavior address a combination of psychological, sociological, and economic concepts to explain what happens in the real economy, in which economic agents present limitations to the exercise of rationality [1,2]. In this regard, various criteria have been used to try to explain the factors that cause biases in financial behavior, highlighting among the most recent ones emotional intelligence (EI) and personality traits (PTs). On the one hand, high levels of EI have been shown to lead to superior employee performance [3–5] and the better financial performance of employees [6]. On the other hand, investor decisions and portfolio performance have also been found to be directly related to PTs [7,8].

For university students, the previous literature on financial behavior shows incipient and controversial results. For example, Haigh and List [9] found that professionals may have stronger financial behavior biases than non-professionals. However, other studies indicate that students have lower EI levels than professionals. Scott-Halsell, Shumate, and Blum [10] concluded that students did not have a sufficient level of EI to be successful leaders, and Scott-Halsell, Blum, and Huffman [11] concluded that there are significant differences between professionals and university students in all areas of EI.

The conclusions obtained from the previous literature indicate that the levels of EI and PTs can influence the financial decisions of both students and professionals, but that these conclusions are still initial and remain controversial regarding the differences

that students and professionals can present. It may be that this controversy is due to data on the psychological influences on financial behavior not being well addressed in student populations [12]. Therefore, it is necessary to understand the abilities and skills of finance students because of their effects on financial behavior will have an important impact on their future personal well-being [13]. Likewise, EI and PTs explain a person's professional achievements, as high reflective skills coupled with high levels of EI are a decisive comparative advantage. However, the understanding of individual determinants of financial success may still be incomplete [14]. The research questions are, then, the following: Is there a difference in EI and PT levels between university finance students and financial industry professionals? If there is, could it cause students to have more financial behavior biases? If differences appear and are related to financial behavior biases, the curricula could incorporate training strategies in EI and PTs with the aim that students begin their careers in the financial industry with the highest levels of non-cognitive skills required to be successful.

To resolve the research questions, EI and PTs were measured from a sample of university students and professionals from the financial industry, confirming the differences between the two samples. These differences between students and professionals were then associated with different biases in financial behavior, which allowed us to determine which aspects of the EI and PTs of students should be used to improve their professional training. From the above, the present work is structured as follows. After this Introduction, a review of the literature is made that relates EI and PTs to the financial behavior of university students. The sample and analysis tools used for the investigation are detailed below. Finally, the results and main conclusions are presented.

## 2. Literature Review

The previous literature on financial behavior has developed different lines of research over time. Initially, financial behavior has been approached from a global perspective, with a descriptive analysis of the different biases in financial behavior [15]. Subsequently, it has tried to find solutions to these biases through two scientific criteria, one of them around the theory of emotions [16–18] and another according to the PTs [2,19–23]. The following is an analysis of the main conclusions obtained by the previous literature on financial behavior forEI, PTs, and university students.

### 2.1. Emotions and Financial Behavior

So far, only a limited number of studies have linked personality psychology and financial behavior. Typically, the main source of inspiration for financial behavior has been cognitive psychology, which offers a broad set of ideas about human decision-making and about the biases that tend to influence decision-making processes [24]. Cognitive psychology proposes that people act based on situations they already know and the interpretation they give them [25]. In the context of this cognitive theory, cognitive biases are defined as those rules that help the individual to simplify complex situations to be able to adopt models of decisions, and convenient actions are identified [26]. Pompian [27] classified biases into two categories: cognitive biases, which involve how people think, and emotional biases, which involve how they feel. For their part, cognitive biases arise from errors in memory and information processing from faulty reasoning, while emotional biases lead to reasoning influenced by feelings. In the area of financial behavior, Pompian [17] points out that investors are subconsciously influenced by past experiences and personal beliefs to the extent that even smart investors can deviate from logic and reason. He considers that EI is related to different biases such as trust in their gut, status quo, loss aversion, self-control, endowment, regret aversion, and affinity. He also identified anchoring bias, associating it with unsophisticated investors who are especially driven by emotional biases that lead them to be especially risk averse.

Munir et al. [18] concluded that people who have control over their emotions can make better investment decisions than those with lower EI, finding a positive and significant

relationship between EI and financial behavior. Ezadinea, Fathi, and Salami [28] also demonstrated the effects of EI on financial behavior biases and investment performance. Pirayesh [29] carried out an empirical study from which it was concluded that there was a relationship between global EI and the investors' decision, pointing out that investors with a high level of EI had a greater riskaversion, understood as mistrust that the individual manifests to accept an offer with an uncertain result compared to another offer with a less profitable but safer result [30]. On the other hand, Tanvir, Sufyan, and Ahsan [31] used Goleman's (1998) five dimensions of EI and demonstrated that there was an influence of global EI and some emotional competencies (emotion regulation, self-motivation, and trait empathy) in the investor decision-making process. Likewise, Dhiman and Raheja [32] stated that individuals who have high values of emotional competencies, emotion regulation, stress management, self-esteem, self-motivation, relationship, and sociability, have a higher level of risk tolerance. In this sense, risk tolerance refers to the amount of risk that the individual is willing to take to obtain a reward [17,33].

### 2.2. PT and Financial Behavior

Van Witteloostuijn and Muehlfeld [34], based on personal psychology and cognitive psychology, concluded that the personality of a human being is a key determinant of his/her performance and behavior, and that contributes to explaining the differences between individuals in terms of susceptibility to financial behavioral biases. From this perspective, an individual's financial decision process is based on a complex combination of demographics and personal characteristics [35]. Demographics include aspects such as age, gender, income, and level of education [36,37], and personal characteristics refer to personality traits, values, and emotions [38,39]. Furthermore, Tauni, Fang, and Yousaf [22] investigated the association between information acquisition and financial behavior by analyzing the influence of the investor's personality. The results of their study indicated that the acquisition of information is directly proportional to the frequency of trading.

Salovey [40] found that there are some PTs, such as extraversion, agreeableness, neuroticism, openness, and responsibility, that can systematically influence investment decisions. Durand, Newby, and Sanghani [7] also demonstrated that investment decisions and portfolio performance are directly related to extraversion, agreeableness, neuroticism, openness, and conscientiousness. Hopfensitz and Wranik [41] associated a greater tendency to yield to risk aversion with high levels of neuroticism. Dhiman and Raheja [32] affirmed that individuals who have high values of extraversion, agreeableness, and openness also have a higher level of risk tolerance. For their part, Durand et al. [21] proposed that investor confidence is associated with PT in two heuristics used to model market movements, that of availability (strategy of estimating the probability of an event occurring through associations that come to mind) and disposition (a tendency to disproportionately hold assets with monetary loss while simultaneously liquidating assets and accrued gains). Other studies have corroborated that financial advice is more likely to increase the frequency of trading when the adviser's personality tends to be frank, conscientious, and personable and that information obtained from financial advisors causes less adjustment in investor portfolios when her/his personality is extraverted and neurotic [42,43]. Furthermore, higher levels of openness could favor investment behaviors in changing markets because new information is more likely to be integrated into decision-making, that is investors with a high level of openness invest less in bad markets because they have greater control over risk averse bias [41].

### 2.3. The Financial Behavior of University Students

Previous studies suggested that EI is an important aspect of academic achievement and student retention [44]. Furthermore, there is a positive relationship between self-control and the employability potential of students [45]. The financial profile of young people of the millennial generation is often characterized by low financial literacy, risky financial behavior, and poor financial satisfaction [46,47]. Due to these generational changes, many

experts consider that a change in the financial education system that affects interpersonal skills such as planning capacity, the confidence to be proactive, and the willingness to take investment risks is also necessary [48,49]. Falahati et al. [13] pointed out that the emotional training of students is important because it influences their financial behavior. Sjöberg and Engelberg [50] indicated that students have a strong tendency towards economic risktaking, achievement orientation, the search for emotions, and a relatively high level of EI. However, they detected a low trend in terms of the value of money, compared to other university students from different branches. Felton, Gibson, and Sanbonmatsu [51] found a relationship between students' optimism and the risk of their investments. Similarly, Saurilin et al. [52], after their research with university students from Portugal and Brazil, concluded that finance students showed less bias toward the status quo in their financial decisions than students without previous studies on finance. Oehler et al. [19] found that extraversion and neuroticism significantly influenced the financial behavior of university students. They pointed out that most extraverted students buy more financial assets when the assets are too expensive and that the more neurotic students have fewer risks in their financial portfolios.

Some studies have also detected the existence of a deficit in the EI of university students with respect to professionals that extends to a large part of the emotional competencies, highlighting among them self-motivation, the ability for emotional expression, and empathy [11,12]. Rzeszutek, Szyszka, and Czerwonka [53] demonstrated that susceptibility to financial behavior biases depends on the level of experience in the financial market and that some PTs such as impulsiveness and empathy are closely related to these biases.

## 3. Methods

### 3.1. Samples

The present study uses two different samples to solve the established research questions. On the one hand, there was a sample made up of a total of 240 students from the degrees of Economics, Business Administration and Management, and Finance and Accounting from a Spanish public university in the academic year 2019/2020, selected from those who had passed at least 80% of the total of the subjects. For this, a request for participation was sent to the population of students who met the characteristics indicated, obtaining a response of 45%. On the other hand, there was a sample made up of a total of 150 professionals selected randomly from among those who carry out their professional activities within the financial industry in the Spanish market in April 2020. Both samples completed EI and PT measurement questionnaires, with the specifications shown in the next section. These questionnaires were previously subjected to a methodological test to detect possible difficulties in their application. In this process, financial experts intervened and provided feedback that modified some questions and the addition of other elements to the final questionnaire. Of the total sample of students, forty-eight-point-zero-four percent were men, and fifty-one-point-nine-six percent were women. The mean age of the students was24.03 years, and fifty-three-point-nine-two percent of them had the intention of starting postgraduate studies at the end of their undergraduate studies. The socio-demographic characteristics of students in the sample appear in Table 1.

**Table 1.** Socio-demographic characteristics of the student sample.

|  | % |
| --- | --- |
| Women | 51.96 |
| Men | 48.04 |
| Average age | 24.03 (years) |
| Postgraduate studies intention | 53.92 |
| Use of scholarship | 67.65 |

In the sample of professionals, sixty percent were men, and forty percent were women, the average age of the participants being 44 years. Furthermore, one-hundred percent of the sample had a university degree, 10% a Ph.D. degree, and 30% a university Master's degree. Likewise, ten percent of the sample had experience in the financial industry of fewer than 10 years, 35% between 11 and 15 years, and 40% of the sample more than 16 years. The socio-demographic characteristics of the professionals in the sample appear in Table 2.

**Table 2.** Socio-demographic characteristics of the professional sample.

|  | % |
|---|---|
| Women | 39.71 |
| Men | 60.29 |
| Average age | 43.55 (years) |
| University degree | 100.00 |
| University Master's degree | 29.82 |
| Ph.D. degree | 9.75 |
| Experience in the financial industry | |
| Less than 10 years | 24.85 |
| Between 11 and 15 years | 35.40 |
| More than 16 years | 39.75 |

*3.2. Instruments*

Both the students and the professionals in the sample completed the EI measurement Trait Emotional Intelligence Questionnaire (TEIQue) [54] and the PT measurement Big Five Inventory (BFI) [55]. EI refers to a person's set of competencies to monitor and understand one's own emotions and those of others and use this affective information to guide one's thoughts and actions. In this study, the emotional competencies that were of particular significance due to their influence on the behavior of students and professionals in previous research were selected [11,56,57]. Furthermore, four dimensions of EI were considered (sociability, emotionality, self-control, and well-being) that grouped 15 emotional competencies [58–60]. The TEIQue version used in this study is the long version, which comprises 153 items, using a Likert-type measurement scale from 1 to 7 (1 = not at all agree and 7 = completely agree). A synthesis of the selected emotional dimensions and competencies appears in Table 3.

**Table 3.** Dimensions and emotional competencies in the emotional intelligence (EI) questionnaire.

| Dimensions | Emotional Competencies | High Scorers See Themselves as |
|---|---|---|
| Sociability | Relationships | Able to maintain satisfactory personal relationships. |
| | Empathy | Able to take another person's perspective. |
| | Emotion perception | About your own and others' feelings. |
| | Emotion expression | Able to communicate their feelings to others. |
| Emotionality | Emotion management | Able to influence the feelings of others. |
| | Social awareness | Connected to superior social skills. |
| | Assertiveness | Frank and ready to defend their rights. |
| Self-control | Emotion regulation | Able to control their emotions. |
| | Adaptability | Flexible and ready to adapt to new conditions. |
| | Impulsiveness (low) | Thoughtful and less likely to give in to their impulses. |
| | Self-motivation | Unlikely to give up in the face of adversity. |
| | Stress management | Able to withstand pressure and regulate stress. |
| Well-being | Happiness | Satisfied with their lives. |
| | Optimism | Likely to "look on the bright side" of life. |
| | Self-esteem | Successful and self-confident. |

For its part, PTs area set of qualities that describe the individual and that can be considered as key drivers of human behavior. In this regard, the BFI questionnaire used consists of 44 items grouped into five traits (extraversion, agreeableness, neuroticism, openness, and conscientiousness) selected from those that were of particular significance due to their influence on financial behavior [7,61]. This questionnaire uses a Likert-type scale of measurement from 1 to 5 (1 = not agree at all and 5 = completely agree). A synthesis of the BFI questionnaire used in the present study appears in Table 4.

**Table 4.** Personality traits (PTs) in the Big Five Inventory (BFI)questionnaire.

| PT | Main Characteristics | Antagonistic Characteristics |
|---|---|---|
| Neuroticism | Anxiety, hostility, depression, shyness. | Calm, secure, relaxed, emotionally strong. |
| Extraversion | Search for emotions, assertiveness, positive emotions, cordiality. | Reserved, withdrawn, shy, lonely. |
| Openness | Emotional, imaginative, idealistic, depth. | Conventional, realistic, traditional. |
| Agreeableness | Helpful, cooperative, compassionate, conciliatory attitude. | Suspicious, individualistic, antagonistic. |
| Conscientiousness | Sense of duty, need for success, impulse control, aimed at accomplishing tasks. | Lazy, purposeless, weak will, careless in moral principles. |

## 4. Results

### 4.1. Descriptive Analysis

Table 5 shows the levels of the emotional competencies of both the sample of students and of professionals. The differences between samples were analyzed using the Mann–Whitney test since the variables related to EI do not follow a normal distribution. The students presented lower levels than professionals in a large part of the emotional competencies, specifically in emotion regulation, relationships, self-esteem, self-motivation, stress management, empathy, optimism, and emotion perception ($p$-value $< 0.05$). However, in social awareness, students outperformed professionals ($p$-value $< 0.05$). Likewise, Table 5 offers the results of the descriptive analysis of the dimensions and the global score of EI. Students presented lower levels than professionals in three of the four EI dimensions (sociability, emotionality, self-control, and well-being) and globally. These results confirm that students have a deficit in EI compared to professionals.

Table 6 presents the results of the descriptive analysis of the PT variables corresponding to the sample of students and professionals. The differences between both samples were analyzed using the $t$-test since the variables related to personality follow a normal distribution. Students presented a lower average in agreeableness, openness, conscientiousness, and extraversion ($p$-value $< 0.05$), but in neuroticism, the average corresponding to students (2.66) was higher than that of professionals (2.39) ($p$-value $< 0.05$). These results also confirm that students presented significant differences with respect to professionals in all PTs, which may explain different behaviors in making financial decisions.

**Table 5.** EI of students and professionals in the sample.

| | Median | | S.D. | | Min. | | Max. | | M-W |
|---|---|---|---|---|---|---|---|---|---|
| | S | P | S | P | S | P | S | P | |
| (A) Emotional competencies | | | | | | | | | |
| Adaptability | 4.38 | 4.46 | 0.71 | 0.58 | 2.15 | 2.85 | 5.69 | 5.54 | 0.09 |
| Assertiveness | 4.14 | 4.43 | 0.72 | 0.77 | 2.71 | 3.14 | 6.14 | 6.29 | 0.14 |
| Emotion expression | 4.25 | 4.42 | 0.82 | 0.67 | 2.42 | 3.25 | 6.00 | 5.83 | 0.08 |
| Emotion management | 4.21 | 4.33 | 0.63 | 0.61 | 2.83 | 2.75 | 5.67 | 5.50 | 0.06 |
| Emotion regulation | 4.40 | 4.53 | 0.68 | 0.72 | 2.67 | 3.00 | 5.73 | 6.00 | 0.03 ** |
| Impulsiveness (low) | 5.00 | 5.00 | 0.83 | 1.08 | 2.60 | 2.80 | 6.60 | 7.00 | 0.65 |
| Relationships | 5.25 | 5.63 | 0.82 | 0.67 | 3.00 | 4.0 | 7.00 | 6.50 | 0.02 ** |
| Self-esteem | 4.76 | 5.24 | 0.76 | 0.64 | 2.29 | 3.47 | 6.41 | 6.53 | 0.00 *** |
| Self-motivation | 4.75 | 5.00 | 0.71 | 0.59 | 2.75 | 3.75 | 6.63 | 6.25 | 0.02 ** |
| Social awareness | 4.67 | 4.33 | 0.63 | 0.55 | 2.83 | 3.00 | 6.33 | 5.67 | 0.03 ** |
| Stress management | 4.43 | 4.64 | 0.61 | 0.38 | 2.43 | 3.86 | 5.64 | 5.57 | 0.03 ** |
| Empathy | 4.33 | 4.56 | 0.52 | 0.57 | 3.11 | 3.44 | 5.33 | 5.67 | 0.04 ** |
| Happiness | 5.88 | 6.13 | 0.91 | 0.72 | 2.88 | 3.88 | 7.00 | 7.00 | 0.13 |
| Optimism | 5.00 | 5.33 | 0.84 | 0.55 | 3.00 | 4.22 | 7.00 | 6.33 | 0.04 ** |
| Emotion management | 4.30 | 4.40 | 0.60 | 0.63 | 2.70 | 2.80 | 5.50 | 6.20 | 0.03 ** |
| (B) EI Dimensions | | | | | | | | | |
| Emotionality | 4.26 | 4.32 | 0.46 | 0.39 | 5.40 | 5.40 | 3.32 | 3.64 | 0.13 |
| Self-control | 4.56 | 4.75 | 0.47 | 0.42 | 5.25 | 5.25 | 2.96 | 3.67 | 0.00 *** |
| Sociability | 4.40 | 4.72 | 0.49 | 0.42 | 5.38 | 5.38 | 3.15 | 4.13 | 0.01 ** |
| Well-being | 5.06 | 5.47 | 0.72 | 0.52 | 6.41 | 6.41 | 2.97 | 3.91 | 0.00 *** |
| (C) EI Global | | | | | | | | | |
| EI global | 4.58 | 4.79 | 0.41 | 0.33 | 5.47 | 5.47 | 3.56 | 4.12 | 0.00 *** |

S: students; P: professionals; M-W: Mann–Whitney test; ***: Sig. at the 0.001 level; **: Sig. at the 0.05 level.

**Table 6.** PTs of students and professionals in the sample.

| PT | Mean | | S.D. | | Min. | | Max. | | *t*-Test |
|---|---|---|---|---|---|---|---|---|---|
| | S | P | S | P | S | P | S | P | |
| Agreeableness | 3.71 | 3.92 | 0.43 | 0.59 | 2.11 | 2.67 | 4.67 | 5.00 | 0.028 ** |
| Openness | 3.35 | 3.59 | 0.64 | 0.52 | 1.60 | 2.40 | 4.70 | 5.00 | 0.015 ** |
| Conscientiousness | 3.60 | 3.89 | 0.58 | 0.50 | 2.00 | 2.67 | 4.67 | 4.78 | 0.002 ** |
| Extraversion | 3.25 | 3.51 | 0.65 | 0.52 | 1.38 | 2.63 | 4.50 | 4.63 | 0.012 ** |
| Neuroticism | 2.66 | 2.39 | 0.75 | 0.62 | 1.38 | 1.00 | 4.38 | 3.75 | 0.033 ** |

S: students; P: professionals; **: Sig. at the 0.05 level.

### 4.2. Impact Analysis

The results obtained in the descriptive analysis indicated that there are significant differences between the EI and PTs of the university students and professionals in the financial industry. Besides, these differences represent a deficit for students compared to professionals in certain emotional skills and PTs. Therefore, the objective of this impact analysis is twofold. Firstly, we tried to identify if the student deficit was related to financial behavioral biases. Secondly, we determined the impact (positive or negative) on the different biases associated with the student deficit. The results of the impact analysis for EI appear in Table 7. Students presented a deficit in self-control compared to professionals. Taking into account the conclusions of previous literature [16], this should lead them to have endowment bias to a greater extent, that is to overvalue their assets and to make incorrect decisions in asset sales. Likewise, students presented a deficit in trait empathy, so they would have more optimism than professionals, reacting in an exaggerated manner to obtain superior performance [17]. Furthermore, the results show that students have deficits in emotion regulation, stress management, self-esteem, self-motivation, and relationships. According to the conclusions obtained by Dhiman and Raheja [32], this indicates that students are less risk tolerant than professionals, which may prevent them from making

aggressive investments to achieve higher returns in the medium to long term. Finally, and referring to global EI, we were also able to observe a deficit of students with respect to professionals. This aspect can lead them to have greater endowment bias, self-control, optimism bias, risk aversion, and regret aversion [29]. On the contrary, and due to this deficit in global EI, students were more risk averse than professionals, so they may tend not to analyze investment risk well.

**Table 7.** Students' EI gaps and impacts.

| Skills, Dimensions, and EI Global | Associated Bias | Sense of Impact |
|:---:|:---:|:---:|
| Self-control | Endowment bias | − |
| Trait empathy | Optimism bias | − |
| Emotion regulation | Risk tolerance | + |
| Stress management | Risk tolerance | + |
| Self-esteem | Risk tolerance | + |
| Self-motivation | Risk tolerance | + |
| Relationships | Risk tolerance | + |
| Social awareness | Risk tolerance | + |
| EI global | Endowment bias | − |
| EI global | Self-control bias | − |
| EI global | Optimism bias | − |
| EI global | Loss-averse | − |
| EI global | Regret aversion | − |
| EI global | Risk-averse | + |

Table 8 shows the results of the impact analysis of the deficit in PTs. Students had lower levels of openness and higher levels of neuroticism than professionals. Then, and as proposed by Hopfensitz and Wranik [41], they can be more risk averse than professionals, resulting in being more likely to make unfavorable investment decisions. Likewise, the results suggest that students have lower levels of openness, extraversion, and agreeableness, which implies that they are less risk tolerant than professionals and, therefore, that they have greater difficulty in building an aggressive portfolio that provides high returns [32].

**Table 8.** Students' PT gaps and impacts.

| PT | Associated Bias | Sense of Impact |
|:---:|:---:|:---:|
| Neuroticism | Risk-averse | − |
| Openness | Risk-averse | − |
| Extraversion | Risk tolerance | + |
| Openness | Risk tolerance | + |
| Agreeableness | Risk tolerance | + |

*4.3. Discussion*

The results of this study are comparable to other findings on students in the United States of America [11] about the deficit in EI presented by the students regarding professionals, in addition to the results of the study by [53], in which they found that experience in the financial market is related to different PTs presented by students and professionals.

The results obtained also indicate that the students in the sample had a deficit in emotion regulation, stress management, self-esteem, self-motivation, and relationships, so they must be less risk tolerant than the professionals. However, these results differ from those obtained by Lin et al. [46] and Felton, Gibson, and Sanbonmatsu [51] when concluding that students have a strong tendency towards economic risk tolerance. Other aspects related to the deficit in student EI and its effects on financial behavior biases have also been highlighted in this study. Such is the case of self-control, empathy, and global EI levels, which are associated with higher endowment bias, optimism bias, self-control bias, loss aversion, and regret aversion, respectively. However, there are no precedents for these results in the previous literature.

On the other hand, regarding the PTs, previous studies showed that impulsiveness, empathy, extraversion, and neuroticism are closely related to the biases of the financial behavior that students present [19,53]. Our results coincide with those previous studies only about the deficit in empathy, but they were not concerned with other PTs such as openness and extraversion, which have not been found in the previous literature, nor the higher levels of neuroticism that students present compared to professionals.

Differences in the levels of EI and PTs can condition the biases of the financial behavior of students with respect to professionals in the financial industry. Despite the significant differences observed, the results obtained show interesting findings for discussion. For example, there were no significant differences in adaptability, assertiveness, emotion expression, emotion management, impulsiveness (low), and happiness. Perhaps this is because a greater personal and work experience in the financial industry does not have the effect of developing certain emotional skills. This is also because university training programs succeed only in certain aspects of EI, pending significant advances in other emotional skills necessary for the better future financial behavior of students in the labor market.

## 5. Conclusions and Implications

The results of the present study confirmed that there are significant differences in EI and PT levels between students and professionals in the financial industry and that such differences are associated with biases in financial behavior. Students present a gap with respect to professionals about relationships, empathy, emotion perception, emotion regulation, self-motivation, stress management, optimism, and self-esteem, as well as in three of the EI dimensions and the global EI scores. Gaps in EI, therefore, indicate that students tend to have optimism bias, risk tolerance, self-control bias, loss aversion, and regret aversion to a greater extent. On the other hand, and as for PTs, students present a gap in agreeableness, conscientiousness, openness, and extraversion and higher scores in neuroticism, confirming that students are more likely to have risk tolerance bias.

Our study presents several contributions to the literature on financial behavior and financial education. First, and from a theoretical perspective, there are significant differences in the levels of EI and PT between university students and professionals in the financial industry that are associated with financial behavioral biases. Previous research has shown that EI and PT influence financial behavioral biases [6] and that university students present a gap in EI compared to professionals from other industries [11]. However, this is the first study to show that university students present a gap in EI and PTs compared to professionals in the financial industry and that this gap is related to some biases of the students' financial behavior. These conclusions open up new research perspectives on EI and PT in the framework of financial behavior and financial education. Likewise, the results obtained confirmed that university students present a gap compared to professionals only in certain emotional competencies and that these are related to optimism bias, risk tolerance, self-control bias, loss aversion, and regret aversion. The literature indicates the existence of other financial behavior biases [15], although there is no empirical evidence that relates them to either EI or PTs.

This study also presents important practical implications for the management of university education and financial companies. Universities should be aware of providing students with full EI and PT training to help them successfully tackle their professional future in the financial industry. Similarly, financial companies should be aware that young professionals have low-risk tolerance and more likely to have endowment bias, optimism bias, self-control bias, and loss aversion. In this way, they should design the management of their investment portfolios taking into account the EI and PT levels of their employees.

Finally, the results of this study suggest future research on financial behavior. First, given that theories of emotions and PTs have only explained some of the financial behavior biases, other studies could check if other theoretical frameworks could explain them. Second, future research should empirically check the financial behavior bias of students

about the deficits of EI and PTs discovered in the present study. Third, future research should address the effects of gender, age, and educational level on the EI and PTs of students and professionals related to the financial industry. Fourth, future research should study what training techniques could serve to correct the EI and PT differences between students and professionals. Last, although some previous work has pointed out that experiential learning techniques have corrected differences in EI, there are still very few that refer to improvements in PTs.

**Author Contributions:** Conceptualization and writing, original draft preparation, A.M.R.-P. and M.T.-D.; investigation, A.M.R.-P. and M.A.F.-G.; methodology and funding acquisition, M.A.F.-G.; supervision, J.M.-G. All authors have read and agreed to the published version of the manuscript.

**Funding:** The authors gratefully acknowledge the financial support of the Chair of Sustainables Economy and Finance (University of Malaga).

**Institutional Review Board Statement:** Not applicable.

**Informed Consent Statement:** Not applicable.

**Data Availability Statement:** Not applicable.

**Conflicts of Interest:** The authors declare no conflict of interest.

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
