# Peer review of "A Study of the Emotional Intelligence and Personality Traits of University Finance Students"

_education, doi:10.3390/educsci11010025_

Round 1
Reviewer 1 Report
Thank you for giving me the opportunity to read and reflect on your paper.
The main drawbacks of the paper are the following:
I suggest considering rewriting the abstract sections–The ideas of the research could have been more effective through the use of elaborative and concise sentences. The abstract as is, does not provide a concise account of the work and conclusion of the research study. It needs to be more structured and synthesized for research clarity.
Introduction is important to provide and elaborate on relevant contextual ideas and background leading to research studies on financial behaviour, and additional literature to lead to some research questions and explain why it is important for this research study.
The hypotheses and/or research questions are missing.
The methodology has many weaknesses.
The section 3.1. Samples is so detailed, I suggest to reduce it.
The findings were not well-presented to readers to understand the focus of the research study.
What is the role of the section 3.2. Instruments for this article?
The Discussion is not well-presented as it does not integrate with the results of the research study to provide coherent scholarly argument.
The Discussion section should connect the research results with relevant literature citations for validity and reliability.
The conclusion is not supported by the research data, which does not indicate a clearer path for future studies on the topic.
The sources are not well presented and discussed. For example, a book - 23. Carley, M., and I. Christie. Managing sustainable development. London: Routledge 2017. – is mentioned only by a simple sentence, p. 2, r. 71-72 “So far, only a limited number of studies have linked personality psychology and behavioral finance [23].”
I suggest to pay attention to English
Reviewer 2 Report
The manuscript is scientifically sound and contains original findings. The idea is interesting and actual for current behavioural studies. Being fulfilled as an interdisciplinary research, combining theoretical background of behavioural theories and their implementation in educational system and employment sphere, the research contributes to the field of study. The research methods are in line with overall purpose. Main findings are obtained based on qualitative research in the relevant students’ and professional sample.
However, some important details should be emphasized and detailed in order to improve the quality of the research before publishing:
- The description of the sample forming as well as the survey carrying require substantial development. At least, this information must include the following: dates of research or time period of the analyzed data; description of the data collection methods (e.g., surveys, interviews, observations); brief summary on the positive and negative methodological experience, e.g., testing of methodological tools, difficulties in the implementation of the initial research concept, discovery of the problems with the methodological tools, unpredictable organizational and methodical situations. It is incomprehensive also how authors have “randomly selected” students “from those who had passed at least 80% of the total of the subjects”. Details should be provided. The same concerns the sample of professionals forming.
- Table 1 contains incorrect formulations (see comments in the text)4
- Authors use abbreviation EI (emotional intelligence) in the Abstract and then in the text. However, in some parts of the manuscript, especially in the final ones, IE appears. Appropriate changes should be done in the whole text.
- Subtitles and some expressions in text contain incorrect statements – behaviour finance, when authors write about financial behaviour.
- The Reference list contains mainly outdated sources. References should be updated considering recent publications in the field, including those indexed in Scopus and WoS. The sources published since 2017 are rare in the Reference list. More comments see in text.
- Technical typo occurs at the bottom of Table 5 – see comment in the text.
Reviewer 3 Report
First, I think this is well structured and well written. It is a good piece.
One point that I would make is that on Tables 5 and 6 there is no mention of the likert-type scale. I do not know, from those tables, what the measure is. So if 'Students on Adaptability' has a 4.38 and the Professionals a 4.46, is that out of 5, 6, or 7? Or 10? I have no idea. Something might be significant at the 0.05 level, but beyond statistical significant, I, as a reader, still have to decide whether a difference of 0.13 in a statistically-significant item at some level is really significant (BTW, in my pdf version, the notes for “**” is cutoff).
That is my only point that could be clarified in the article.
Reviewer 4 Report
The authors collect a variety of emotional intelligence (EI) and personality traits (PT) from university finance students and finance professionals. They compare the two groups on these measures and then speculate upon the implications for preparing finance students for their professions, particularly on the issue of financial cognitive biases. While the topic of EI and PT development among finance students and professionals is a novel idea, I have several significant concerns with the manuscript. I hope the authors will find them useful as they continue this work.
1) My main critique of the paper is a lack of theory and attention to carefully generating hypotheses. The paper is an atheoretical empirical exercise where the authors measures EI and PT among respondents. Although there is plenty of attention to why EI and PT are important (prior work documents some professional benefit to having more of these skills/abilities), there is no attention to describing (1) why someone should expect differences between professionals and students and (2) how each measure of PI or ET is hypothesized to be correlated with various forms of financial cognitive biases. If anything, there is only a single line about theory on page 7 – near the end of the manuscript – where the authors make some post-hoc rationalization to discuss implications of their findings: “Students present a deficit in self-control compared to professionals. This should lead them to commit endowment bias to a greater extent, that is, to overvalue their assets and to make incorrect decisions in asset sales.” In the literature review, the authors need to discuss issues similar to the one raised in this sentence, namely, what is the theoretical reason why and how each aspect of EI and PT are correlated with cognitive biases and reasons for expecting differences between students and professionals. A discussion of this theory should then lead to some hypotheses that the authors can then formally test. To add another empirical concern, without this disciplined discussion of theory and hypotheses testing, I cannot confidently discern between what findings might be false positives or false negatives.
2) The other major shortcoming in this paper is that the authors discuss at length the connection between EI and PT with various forms of cognitive bias. However, the entire discussion is speculative because the authors never measure cognitive bias, as far as I can tell. If the authors want to connect EI and PT to cognitive bias, they must measure cognitive bias and analyze correlations to make some empirical case for their claims. Currently, they merely draw upon prior literature and make conjectures (summarized Table 7), which is insufficient for making the conclusions they wish to make about the importance of educating for PI and EI in order to address cognitive bias.
3) I also think the authors need to better defend the claim that the differences they observed between professionals and students is due to their status as professionals and students. That is, how can we rule out other potential explanations for these differences? For instance, it is possible that differences in gender composition, age, or even educational attainment between the two groups is the explanation for the differences in EI and PT? There is a literature documenting how EI and PT differs by gender and age. Moreover, there are plenty of other unobservable differences between students and professionals. In particular, not all finance students become finance professionals; there is systematic selection into a professional finance job. Does this selection explain differences in EI and PT between students and professionals? If the authors are detecting differences attributable to these other factors, it is unclear how effective higher education can be in addressing the deficits that the authors claim to exist.
Reviewer 5 Report
This study investigates the impact of IE and PT on the financial behaviour of university students in comparison to financial professions in Spain. It is generally well written. Here I want to point out some aspects that the authors may consider to revise or improve.
1) The title of the paper is "A study of the non-cognitive skills of university finance students". It does not mention directly the two key dimensions measured in the study: EI and PT. Moreover, the overall research design is a comparative study, where the responses of financial professionals are highly relevant. Therefore, it would be more ideal to indicate such key variables or dimensions.
2). In 3.2 Instruments section, the Trait Emotional Intelligence Questionnaire and Big Five Inventory are used as the major research instruments. But what are they? The authors should not assume that all readers are familiar with them. A few lines to explain them would be necessary.
3) In the results part, the descriptions of the results in the tables are too short. There are only a couple of lines to present the data featured in each table, which is not detailed enough to showcase the main findings.
Round 2
Reviewer 1 Report
The paper was improved, but it needs improvement.
The methodology has no logical flow. This is a main weaknesses of the paper.
See my recommendations from previous evaluation and try to find some research questions or hypotheses related to the main objective of the article.
Good luck!
Reviewer 2 Report
Considering the fulfilled corrections, the paper is acceptable for publication
Reviewer 5 Report
Compared to the original manuscript, the current version has shown major revisions in the research presentations, engaging with the comments and suggestions from the peer reviewers.
Round 3
Reviewer 1 Report
The paper was improved! Good luck!